# Association of P10L Polymorphism in Melanopsin Gene with Chronic Insomnia in Mexicans

**DOI:** 10.3390/ijerph18020571

**Published:** 2021-01-12

**Authors:** Bianca Ethel Gutiérrez-Amavizca, Ernesto Prado Montes de Oca, Jaime Paul Gutiérrez-Amavizca, Oscar David Castro, Cesar Heriberto Ruíz-Marquez, Kricel Perez Conde-Andreu, Ricardo Pérez Calderón, Marisela Aguirre Ramírez, Jorge Alberto Pérez-León

**Affiliations:** 1Chemical Biological Sciences PhD Graduate Program, Department of Chemical Sciences, Biomedical Sciences Institute, Ciudad Juarez Autonomous University, Chihuahua 32310, Mexico; ethel90210@gmail.com (B.E.G.-A.); jaime.gutierrez@uacj.mx (J.P.G.-A.); al114657@alumnos.uacj.mx (O.D.C.); al97587@alumnos.uacj.mx (C.H.R.-M.); kandreu92@gmail.com (K.P.C.-A.); marisela.aguirre@uacj.mx (M.A.R.); 2Laboratory of Regulatory SNPs and Laboratory of Pharmacogenomics and Preventive Medicine, Personalized Medicine National Laboratory (LAMPER), Pharmaceutical and Medical Biotechnology, Central Unit, CIATEJ, A.C., National Council of Science and Technology (CONACYT), Guadalajara 44270, Mexico; eprado@ciatej.mx or; 3Scripps Research Translational Institute & Scripps Integrative Structural and Computational Biology Research Institute La Jolla, La Jolla, CA 92307, USA; 4Genomics Sciences Masters Program, Department of Chemical Sciences, Biomedical Sciences Institute, Ciudad Juarez Autonomous University, Chihuahua 32310, Mexico; al154318@alumnos.uacj.mx; 5Cuerpo Académico Consolidado Biología Celular y Molecular, Instituto de Ciencias Biomédicas, Universidad Autónoma de Ciudad Juárez, Ciudad Juárez, Chihuahua 32310, Mexico

**Keywords:** *OPN4* gene, melanopsin, insomnia, SNPs, retina, chronotypes

## Abstract

The aim of this pilot study was to determine the association of the P10L (rs2675703) polymorphism of the *OPN4* gene with chronic insomnia in uncertain etiology in a Mexican population. A case control study was performed including 98 healthy subjects and 29 individuals with chronic insomnia not related to mental disorders, medical condition, medication or substance abuse. Samples were genotyped by polymerase chain reaction-restriction fragment length polymorphism (PCR-RFLP). Genetic analyses showed that the T allele of P10L increased risk to chronic insomnia in a dominant model (*p* = 1 ×10^−4^; odds ratio (OR) = 9.37, CI = 8.18–335.66, Kelsey statistical power (KSP) = 99.9%), and in a recessive model (*p* = 7.5 × 10^−5^, OR = 9.37, KSP = 99.3%, CI = 2.7–34.29). In the insomnia group, we did not find a correlation between genotypes and chronotype (*p* = 0.219 Fisher’s exact test), severity of chronic insomnia using ISI score (*p* = 0.082 Fisher’s exact test) and ESS score (*p* ˃ 0.999 Fisher’s exact test). However, evening chronotype was correlated to daytime sleepiness severity, individuals with an eveningness chronotype had more severe drowsiness according to their insomnia severity index (ISI) score (*p* = 0.021 Fisher’s exact test) and Epworth sleepiness scale (ESS) score (*p* = 0.015 Fisher’s exact test) than the morningness and intermediate chronotype. We demonstrated that the T allele of the P10L polymorphism in the *OPN4* gene is associated with chronic insomnia in Mexicans. We suggest the need to conduct larger studies in different ethnic populations to test the probable association and function of P10L and other SNPs in the *OPN4* gene and in the onset of chronic insomnia.

## 1. Introduction

Insomnia is characterized by the difficulty to initiate and/or maintain sleep, which results in either chronically non-restorative or poor quality sleep. It is a serious health problem with a prevalence of 10–50% in the general population. Patients with insomnia have a positive family history of ~35% [1]. Many factors are involved in insomnia—environmental as well as genetic—even though these components have not been clearly defined. The global literature contains only one case report of a patient with chronic insomnia, which was a heterozygous carrier of nonsense mutation in the *GABRB3* gene. This mutation is located in exon six and consists of the substitution of the amino acid arginine with histidine at position 192 (R192H). The patient had a family history of sleep problems [2].

Melanopsin is a photosensitive protein present in a subset of the ganglion cells in the retina of all vertebrates, including humans. The melanopsin-containing cells are known as intrinsically photosensitive retinal ganglion cells (ipPRG). ipPRG act as an additional photoreceptor to the classical rod and cone cells—reviewed in [3]. They directly project into the suprachiasmatic nuclei (SCN), the pacemaker of biological rhythms. Melanopsin mediated phototransduction, performed by the ipRGCs, is complementary to the activity of cones and rods for SCN photosynchronization, but only the ipPRG axons constitute the retinohypothalaic tract that conducts the nerve impulse required for SCN photosynchronization [3]. It is then understandable that a large number of investigations have focused on the relationship of melanopsin and ipPRG in behavioral disorders of the circadian rhythm—reviewed in [4]). In humans, the participation of melanopsin has already been demonstrated in different behavioral disorders, such as seasonal affective disorder [5] and delayed sleep/wake timing [6] (see below).

There are three isoforms of protein opsin 4 consisting of 478, 489 and 495 amino acid residues, they have a molecular weight of 52.6, 53.8 and 54.5 kDa [7]. Opsin 4 is encoded by the *OPN4* gene, located on chromosome 10q23.2, with a length of 12,292 bases (Genecards). Functional single nucleotide polymorphisms (SNPs) identified in the *OPN4* gene are: P10L (rs2675703), I394T (rs1079610), and D444G (rs12262894). See Appendix A. The P10L polymorphism is a change from proline to a leucine at position 10, it seems to not interfere with the correct distribution of melanopsin on the plasma membrane. However, SNP P10L has been associated with seasonal affective disorder, a condition where depressive states are experienced in autumn and winter, when days are shorter [8,9]. The I394T polymorphism is a change of isoleucine by tyrosine at position 394 and has been associated with a pupillary light response under various photic conditions including different intensities and wavelengths [10]. The D444G polymorphism consists of the replacement of aspartic acid by glycine at position 444, it is frequent in African populations (4.9%, and has been classified as a benign variant with no associated phenotype [11]. These different associations of the *OPN4* gene SNPs show how melanopsin mediated photransduction and ipRGCs activity are associated with several light-driven functions not related to vision.

Affective and sleep disorders are related. Correct sleep homeostasis is dependent upon an efficient sensing of environmental light. The P10L SNP associated with affective disorder, could be also linked to a defective sensing of environmental light leading to perturbance in sleep homeostasis, manifesting most severely as difficulty initiating and/or maintaining this process. Hence, we wanted to seek an association between the P10L polymorphism *OPN4* gene and insomnia. Thus, the aim of this study was to determine whether there was an association between the *OPN4* gene P10L polymorphism and chronic insomnia of uncertain etiology in a Mexican population.

## 2. Materials and Methods

### 2.1. Subjects

We included 98 healthy individuals as a control group. This consisted of individuals who were older than 18 years, who did not have family history or suffer from a primary sleep disorder. The patient group included 29 individuals diagnosed with chronic insomnia of uncertain etiology according to the criteria of the Diagnostic and Statistical Manual of Mental Disorders (DSM–5). Excluding criteria were a history of mental disorder, medical condition, medication or substance abuse. All patients and healthy subjects were provided with written informed consent to participate in this study, in adherence with the Declaration of Helsinki and General Health Law of Mexico that regulates health and research. This study was accepted by the Institutional Committee of Ethics and Bioethics of the University Autonomous of Juarez City (CIBE-2016-1-40).

The control group participants were required to self-complete a set of questionnaires including clinical sleep characteristics and chronotype evaluation. The Composite Scale of Morningness (CSM)—adapted in Spanish [12]—and the chronic insomnia group participants answered the same questionnaires as the control group, plus the Epworth Sleepiness Scale (ESS), insomnia severity index (ISI) and Pittsburgh scale quality index (PSQI).

### 2.2. Instruments

#### 2.2.1. Chronotype Evaluation Composite Scale of Morningness (CSM)

The CSM was used to categorize chronotype in one of three groups: morningness, intermediate and eveningness [13].

#### 2.2.2. Epworth Sleepiness Scale (ESS)

The ESS ranks daytime sleepiness into: daytime sleepiness; fair likely daytime sleepiness; highly daytime sleepiness; and excessive daytime sleepiness [14].

#### 2.2.3. Insomnia Severity Index (ISI)

The ISI assesses the severity of both nighttime and daytime components of insomnia and grades into: no clinically significant insomnia, moderate clinical insomnia and severe clinical insomnia—see details in references [15,16].

#### 2.2.4. Pittsburgh Scale Quality Index (PSQI)

The PSQI evaluates sleep quality and disturbances along the previous month. A score >5 suggests poor sleep quality [17].

### 2.3. Genotyping

Venous blood was collected in EDTA tubes and genomic DNA was extracted according to Miller [18]. SNP rs2675703 (P10L) was genotyped using an RFLP-PCR assay modified from [5]. Primer sequences were: forward 3′-AGGAAAGTTGGGAGGCTGAG-5′, reverse 3′-GGTCAGGGAAGGCTCTGTG-5′. End-point PCR of 100 ng of genomic DNA was performed with the following conditions: denaturation for 10 min at 95 °C; followed by 45 cycles of 30 s at 94 °C (denaturation), 1 min at 57 °C (annealing), and 1 min at 72 °C (elongation); a final elongation step of 10 min at 72 °C. A unique 281 bp amplicon of the *OPN4* gene with no primer-dimers or unspecific amplicons was obtained in all assays. The PCR products were digested with an *Aci*I restriction enzyme at 37 °C for 6 h. The designed assay was possible due to two previous observations: (i) rs2675703 (P10L) is adjacent to a second SNP rs11202106 and recombination between these two markers does not occur (D’ = 1 according to LDlink [19] and also suggested by Mexicans in Los Angeles (MXL) subpopulation from 1000 genomes project (1kGP, C allele frequency of rs2675703 = 0.875 and G allele of rs2675703 = 0.867; (ii) there are six possible haplotypes in the world population (ALL populations from 1kGP) including CG, TA, CA, CC, TG and CC. SNP rs11202106 is biallelic and not triallelic in MXL subpopulation (C allele in rs11202106 does not exist). Thus, the *Aci*I cut haplotype CG (allele C of interest) and does not cut in haplotype TA (allele T of interest). So, the T/T genotype generated a unique undigested 281-bp product, the C/T genotype produced three bands of 281 bp, 161 and 120 bp, and the C/C genotype yielded two fragments of 161 and 120.bp. The digested amplicons were analyzed using electrophoresis in 2% agarose gels, followed by silver staining.

### 2.4. Statistical Analysis

Allele frequencies were determined by counting, and the distribution of genotypes in both groups was compared using the χ^2^ test or Fisher’s exact test. The Hardy–Weinberg equilibrium was calculated in controls using the χ^2^ test. Odds ratios (ORs) with 95% confidence intervals (95% CIs) were obtained to estimate the associations between genotypes and disease in a codominant model according to Lewis [20]. A *p*-value of <0.05 was considered statistically significant. Kelsey and Fleiss parameters were calculated as post hoc statistical powers for each genotype comparison with EpiInfo software (StatCalc, Centers for Diseases Control, Atlanta, GA, USA) v. 7.2.4.0. Two-sided confidence intervals of 95% was considered in all genetic models.

## 3. Results

### 3.1. Clinical Characteristics Related to Sleep

We attempted this study trying to find an association between the polymorphism P10L of the *OPN4* gene and chronic insomnia. Subjects were included in two groups: those with no difficulty initiating or maintaining sleep were considered as healthy (control group, *N* = 98, 56 men, 42 women) and individuals suffering from chronic insomnia (*N* = 29, 20 women, 9 men). The median age in the control group was 23.14 ± 6.36 (mean ± SD) (range 17–61), while in the insomnia group it was 38.62 ± 13.85 (mean ± SD) (range 18–63). In the insomnia group, an obesity trend was identified, defined as body mass index (BMI) [21] overweight in 44.8%, BMI between 25–29.9, type 1 obesity BMI between 30–34.9, in 13.8%, and type 2 obesity in 6.9% (BMI between 35–39.9). Whereas this distribution in the control group was restricted to normal weight 59.6%, overweight 24.6% and type 1 obesity 9.7%. with no type 2 obesity. These differences were significant (Fisher’s exact test, *p* < 0.012).

Sleep duration in hours in the insomnia group was 5.41 ± 1.34 (mean ± SD) (range 3–9), however, it must be taken into account that 62% were taking medication for insomnia. No significant differences were found between groups in the chronotype (*p* = 0.241).

Clinical features, symptoms and behavioral disturbances related to sleep such as apnea, bad quality of sleep, lacking feeling of restful sleep, difficulty in maintaining sleep, difficulty in concentration, alteration in mood, muscular tension, seasonal depression and anxiety were statistically significant, being more prevalent in the insomnia group (Table 1). Patients with chronic insomnia reported poor sleep quality in 62.1%, while in the control group it was only the 11%. However, when sleep quality was measured in diurnal naps no differences were found. A total of 89.7% of patients with insomnia reported a poor feeling of rest, while in the control group it was 42.2%.

Family history of at least one member with insomnia in individuals with chronic insomnia was found in 58.6%. Moderate clinical insomnia according to the severity index was found in 37.9% of individuals and in 24.1% this was clinically severe. According to Pittsburgh scale, all individuals (100%) with insomnia were bad sleepers. Age of onset of symptoms in chronic insomnia individuals was 26.86 ± 14.51 (range 10–60). Conforming to EPS, individuals were distributed according to classification: unlikely (51.7%), fairly likely (13.8%) highly likely (13.8%) and excessively likely (20.7%)**.**

### 3.2. Genotyping, Allelic Frequencies and Correlation to Sleep

Genotype and allele frequencies of the rs2675703 (P10L) polymorphism of the *OPN4* gene are shown in Table 2. There is a strong association between the occurrence of the *OPN4* gene P10L SNP and insomnia, genotype distributions in the control group were in agreement with the Hardy–Weinberg equilibrium. In the chronic insomnia group, we did not find a correlation between the genotype and chronotype (*p* = 0.219 Fisher’s exact test), daytime sleepiness severity using ISI score (*p* = 0.082 Fisher’s exact test) or ESS score (*p* ≥ 0.999 Fisher’s exact test), but when we analyzed the correlation between chronotype with daytime sleepiness severity we found that individuals with an eveningness chronotype had more severe drowsiness according to ISI score (*p* = 0.021 Fisher’s exact test) and ESS score (*p* = 0.015 Fisher’s exact test) than the morningness and intermediate chronotype (Table 3).

## 4. Discussion

Chronic insomnia is the most common primary sleep disorder in the population. Among the risk factors are advanced age, female sex, stress, substance abuse, as well as medical and psychiatric illnesses. There are few genetic studies regarding genetic variations involved in the susceptibility to insomnia. Genetic information can contribute to our understanding of its pathophysiology, the screening of risk factors and the development of an effective treatment. The participation of the ipPRG in the regulation of the sleep wake cycle and therefore the participation of the *OPN4* in the integrity of this cycle is evident. Table 4 shows a brief summary of studies reported to date on the genetic contribution of *OPN4* gene polymorphisms to sleep and their disturbances which corroborate the implication of melanopsin in the regulation of the photo response, chronotype and sleep cycles. Noteworthy, one study was performed in a Japanese population and two more performed in Caucasian populations, thus, our study is the first report analyzing *OPN4* SNPs in a Hispanic/Latin American population sample.

We found no differences between chronotypes of patients when compared with chronotypes of controls. To our knowledge, no previous study has attempted to sort patients suffering with chronic insomnia into chronotypes, namely, whether individuals could be classified as morning (morningness), intermediate, or evening (eveningness) types. As surprising as this is, for us, the results of particular noteworthiness—regardless of the difficulty to initiate/maintain the sleep—individuals still show a diurnal preference similar to control group.

We demonstrated several statistically different behavioral disturbances in the insomnia group when compared with the control group. From Table 1, the data evidently show that insomnia patients are more susceptible to apnea, bad quality of sleep, lack the feeling of having a restful sleep, difficulty in concentration, alteration in mood, muscular tension, seasonal depression and anxiety. In this way, our analysis reports additional features that lead to a lower quality of life in insomnia patients.

Furthermore, we also detected a trend for the insomnia group to present overweight and obesity: almost one half of the patients showed overweight (44.8%), and considering both type 1 and type 2 altogether, 20.7% of individuals were obese (13.8% and 6.9%, respectively). That makes a total proportion of more than 65% patients of chronic insomnia suffering some metabolic disfunction. According to this, it has been demonstrated that the insomnia phenotype is associated with a significant risk of cardiometabolic and neurocognitive morbidity and mortality [22]. Additionally, several studies have demonstrated a strong association between persistent complaints of difficulty initiating or maintaining sleep with an increased risk of hypertension [23], acute myocardial infarction [24] and type 2 diabetes [25].

We performed the genotyping of control and insomnia subjects seeking for the occurrence of *OPN4* alleles variants. Our analysis regarding alleles and genotypes strongly reveals a higher risk of insomnia when the T allele is present in all the genetic models tested. The contribution of allele T was statistically significant (*p* ≤ 0.001, OR = 9.73, CI = 4.78–19.93), suggesting an almost ten-fold risk of occurrence of insomnia in individuals with the T allele. Therefore, the T allele is a risk factor for chronic insomnia, based on all three inherited models including dominant (*p* ≤ 0.001, OR = 37.38, CI = 8.18–335.66), recessive (*p* = 7.5 × 10^−5^, OR = 9.37 CI = 2.70–34.29) and codominant (*p* = 7.12 × 10^−4^, OR = 4.8, CI = 1.8–12.71). It is important to highlight that when the genotype TT is present, this conveys a risk of being 37.38 times more likely to suffer from chronic insomnia.

Additionally, there is no correlation between genotypes with chronotype and daytime sleepness severity, but interestingly, the chronotype was correlated with daytime sleepiness severity. Individuals with an eveningness chronotype had worse prognosis in terms of severity and sleepiness during the day than those with a morning or intermediate chronotype. Is well known that an evening chronotype carries negative consequences for sleep and reports of higher levels of daytime sleepiness [26] and more maladaptive beliefs about sleep compared to morning types [27]. We found that the chronic insomnia group has >68% more patients with the eveningness chronotype than controls. It is probable that the chronotype shift towards eveningness could be a pre-insomnia state, if insomnia is seen as a threshold disease. If this is true, this result would lead to an opportunity to prevent insomnia cases in future approaches. However, this remains to be demonstrated in a longitudinal and larger study. We demonstrated that in all the genotype and allele comparisons the statistical power was above 94.3%, in spite of a low sample size. The high statistical power in our result can be explained due to (i) the marked differences in allele and genotype frequencies (exposures) between cases and controls and to (ii) and the high control/case ratio (3.38).

Evening chronotype has also been associated with clinical depression [28,29], psychopathology in adolescents [30], bipolar disorder [31], subclinical mania [32], and insomnia [33]. Taken together, these studies and ours suggest that poor sleep and eveningness could be a predictive marker of insomnia severity. Therefore, insomnia patients with an evening chronotype are particularly high-risk group for sleepiness severity and susceptibility of other mental disorders.

A similar association of this SNP has been found for patients suffering with seasonal affective disorder [5], and also for healthy individuals in predicting their sleep timing and chronotype: earlier bedtimes in shorter days and later bedtimes in longer days [6]. There is no correlation so far between the substitution of proline 10 by leucine in the melanopsin protein and the alteration of its function, even more, it seems that the peptide with the substitution has normal sorting to the plasma membrane [6]. It is therefore difficult to hypothesize on a feasible role for this allele variation leading to a disturbance in light perception and thus in the etiology of insomnia, joined to the fact that P10L did not show melanopsin loss-of function in vitro, with no evidence of protein misfolding in cellular localization. The only reported effect is that cells expressing P10L showed a reduced amplitude response compared with the wild type [34]. Proline to leucine mutations in transmembrane proteins can lead to altered chaperone-mediated stabilization and proteasome-dependent degradation in the central nervous system [35]. Another probable explanation is that P10L could be in linkage with a functional variant. In this regard, additional in silico evidence suggests that P10L SNP (rs2675703) is in complete linkage with the intron SNP rs2254051, having both the same population frequency of 12.5%, as is expected according to 1kGP in Mexicans in the Los Angeles population. However, SNP rs2254051 has no predicted functional effect nor does it overlap with any regulatory features (See Appendix A, [36])

Regardless of the extensively documented role of melanopsin and ipPRG with photoentrainment, there are no studies on the association of the P10L melanopsin SNP with sleep disorders, beyond Seasonal affective disorder (SAD), sleep/awakening time and chronotype reported by Rocklein et al., [5,37] and Lee et al. [6]. Hence, we do not have a source of comparison for our data, nevertheless, it is clear that we have identified an association for P10L with insomnia. Our results suggest that melanopsin is involved in sleep disorders, at least in chronic insomnia. We encourage research groups to conduct larger studies in populations with different ethnic backgrounds in order to reinforce this hypothesis.

## 5. Conclusions

Melanopsin and ipPRG have a primary role in non-visual retinal light responses. Thus, an alteration of melanopsin signaling or ipPRG signaling to encephalic nuclei could also lead to disturbances in sleep control. Several single nucleotide polymorphisms have been reported for the *OPN4* gene, and some of these have been related to alterations in extravisual light responses. We analyzed—by genotyping—a group of insomnia patients to look for an association of the P10L polymorphism *OPN4* gene. We found a higher risk of suffering with insomnia to be associated with the occurrence of this allele variant. Our preliminary results add evidence to the role of melanopsin and ipPRG in the control of light driven behaviors involved in the sleep–awake cycle, but further research is needed to demonstrate the biological impact of this SNP or its associated functional variant.

## Figures and Tables

**Table 1 ijerph-18-00571-t001:** Clinical characteristic and behavioral disturbances related to sleep in chronic insomnia patients.

Characteristics	Control Group *N* = 98 (%)	Chronic Insomnia Group *N* = 29 (%)	*p*-Value
**Apnea**			
Yes	9 (9.2)	15 (51.7)	
No	89 (90.8)	14 (48.3)	<0.001
**Sleepiness**			
Yes	63 (70.4)	23 (79.3)	
No	34 (34.6)	6 (20.7)	0.364
**Take a nap during day**			
Yes	39 (39.8)	7 (24.1)	
No	59 (60.2)	22 (75.9)	0.186
**Sleep quality**			
Good	30 (30.6)	0	
Regular	58 (59.2)	7 (24.1)	
Bad	9(9.2)	18 (62.1)	* <0.001
**Sensation of restful sleep**			
Yes	51 (52)	3 (10.3)	
No	40 (40.2)	26 (89.7)	* 0.001
**Somnambulism**			
Yes	10 (10.2)	3 (10.3)	
No	87 (88.8)	26 (89.7)	˃0.999
**Night terrors**			
Yes	13 (12.9)	2 (6.9)	
No	83 (83.6)	27 (93.1)	0.713
**Nightmares**			
Yes	30 (30.6)	11 (37.9)	
No	68(69.4)	18 (62.1)	0.501
**Difficulty maintaining sleep**			
Yes	22 (22.4)	25 (86.2)	
No	76 (77.6)	4 (13.8)	* <0.001
**Fatigue**			
Yes	70 (71.4)	25 (86.2)	
No	28 (28.6)	4 (13.8)	* 0.145
**Bad performance school/work**			
Yes	44 (44.9)	17 (58.6)	
No	54 (55.1)	12 (41.4)	0.211
**Difficulty concentrating**			
Yes	50 (51.0)	22 (75.9)	
No	47 (48.0)	7 (24.1)	0.036
**Difficulty memorizing**			
Yes	39 (39.8)	16 (55.2)	
No	59 (60.2)	13 (44.8)	0.200
**Mood alteration**			
Yes	34 (34.7)	24(82.8)	
No	62 (63.3)	5 (17.2)	0.002
**Muscular tension**			
Yes	44 (44.9)	21 (72.4)	
No	53 (541)	8 (27.6)	0.016
**Headache**			
Yes	33 (33.7)	15 (51.7)	
No	64 (34.7)	14 (48.3)	0.184
**Seasonal depression**			
Yes	14 (14.3)	16 (55.2)	
No	83 (84.7)	13 (44.8)	0.001
**Anxiety**			
Yes	39 (39.8)	22 (75.9)	
No	59 (60.2)	7 (24.1)	0.001

χ^2^ test, significative *p*-value < 0.05, * Fisher’s test when expected cell is less than 5.

**Table 2 ijerph-18-00571-t002:** Genotype and allelic frequencies of *OPN4* in chronic insomnia.

Genotype	Controls	Insomnia Patients	*p*-Value, OR (95% CI)	Statistical Power	
				(Kelsey)	(Fleiss)
	*N* = 98 (%)	*N* = 29 (%)			
C/C	72(79.6)	2(6.9)			
C/T	20(20.4)	16(55.2)			
T/T	6(7.0)	11(37.9)			
Model					
Dominant					
TT + CT	26	27			
CC	72	2	*p* = 1 × 10^−4^, 37.38 (8.18–335.66)	99.9995%	99.99999999985%
Recessive					
TT	6	11			
CC + CT	92	18	*p* = 7.5 × 10^−5^, 9.37 (2.70–34.29)	99.3%	97.0%
Codominant					
CT	20	16			
CC + TT	78	13	*p* = 7.12 × 10^−4^, 4.8 (1.8–12.71)	95.4%	94.3%
Allele					
C	164	20			
T	32	38	*p* = 0.0001, 9.73 (4.78–19.93)	99.94%	99.93%

**Table 3 ijerph-18-00571-t003:** Correlation between chronotype and insomnia severity according to insomnia severity index (ISI) and Epworth Sleepiness Score (ESS).

Chronotype	ISI Score		*p*-Value	ESS Score		*p*-Value
	Subclinical	Moderate-Severe		Unlikely Sleepiness	Likely Sleepiness	
Morningness	5 (50%)	2 (50%)		6 (85.7%)	1 (14.3%)	
Intermediate	5 (15.4%)	6 (84.6%)		9 (81.8%)		
Eveningness	1 (60%)	10 (40%)	0.021	3 (27.3%)	8 (72.7%)	0.015
Fisher’s exact test					Fisher’s exact test	

**Table 4 ijerph-18-00571-t004:** Genetic contribution to P10L polymorphism melanopsin gene.

Polymorphism	Characteristic	*N*	OR (95%CI)/F Test	*p*-Value	Population	Reference
P10L rs2675703 c.29C>T Pro10Leu	Seasonal affective disorder (SAD)	130 SAD 90 control	5.63 (1.22–26.01)	<0.05	95% Non-hispanic Caucasic	Roecklein et al., 2009
	Sleep onset Chronotype	234 healthy individuals	F(9230) = 2.469 F(2229) = 3.214	<0.05 <0.05	Non-Hispanic Caucasic	Roecklein et al., 2012
	Bed time Awakening time	348 healthy individuals	F = 7.058 F = 3.353	<0.01 <0.05	Japanese	Lee et al., 2014
	Chronic insomnia	98 controls 29 chronic insomnia	37.38 (8.18–335.66)	<0.01	Mexican	This study

## Data Availability

The data presented in this study are available on request from the corresponding author.

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
