# Peer review of "Association of P10L Polymorphism in Melanopsin Gene with Chronic Insomnia in Mexicans"

_ijerph, 2021, doi:10.3390/ijerph18020571_

Round 1
Reviewer 1 Report
Comments
Although the meaning of sentences can be gleaned, the manuscript would benefit from thorough editing from someone with full professional proficiency in English. Some examples that I found difficult to interpret include lines 43, 155-156, and 203.
Line 31. Why are the confidence intervals different (99.3% and 95.4%)? It would be better to state the 95% Cis for each.
Line 34. I don’t think that we can state with confidence that a p-value is 1 (just as we can’t state with confidence that it is 0). You might want to state p>.999 instead (unless journal style says otherwise).
Line 52. The accepted abbreviation is ipRGC (intrinsically photosensitive retinal ganglion cell). This abbreviation should be used throughout the manuscript.
Line 54 & 55. Citations are needed for each of these claims.
Line 59. “Behavioral disorders” is vague, can you list them instead?
Line 60. The link between OPN4 and ipRGCs is not explicitly stated and so the relevance is unclear to the non-expert.
Line 62-65. Some citations are needed.
Line 69. What is known about D444G, functionally?
Line 70 - 73. The logic here is not well articulated. ipRGCs sense light for circadian rhythms (process C), but you are investigating insomnia (usually thought to be a disturbance of the sleep homeostat, process S). Can a more thorough argument be made? This is important because the scientific rationale for your study is not clear at the moment.
Line 159. I don’t know what type 1 and type 2 obesity is. Is it linked to diabetes status? The authors should briefly explain.
Table 1. I’m not sure what the numbers in brackets denote. Also, where does the n=114 come from?
Line 201. Doesn’t citation 27 discuss this?
Author Response
REVIEWER 1 COMMENTS
We uploaded the track changes word document : P10L_ijerph-1052002ms_VersionafterReviewersTrackChnges, this file does not follow the original line numbering, please see the file: P10L_ijerph_1052002ms_VersionafterReviewers with the changes incorporated to follow the original line numbering
1.- Although the meaning of sentences can be gleaned, the manuscript would benefit from thorough editing from someone with full professional proficiency in English. Some examples that I found difficult to interpret include lines 43, 155-156, and 203.:
The grammar of the entire manuscript has been revised with particular attention to the suggested lines by the reviewer.
2.- Line 31. Why are the confidence intervals different (99.3% and 95.4%)? It would be better to state the 95% Cis for each.
These percentages refer to statistical power, now this has been specified in the abstract to avoid confusion to the readers.
3.- Line 34. I don’t think that we can state with confidence that a p-value is 1 (just as we can’t state with confidence that it is 0). You might want to state p>.999 instead (unless journal style says otherwise).
This has been corrected in both table 1 and abstract according to reviewer´s recommendation.
4.- Line 52. The accepted abbreviation is ipRGC (intrinsically photosensitive retinal ganglion cell). This abbreviation should be used throughout the manuscript. Now is corrected.
5.- Line 54 & 55. Citations are needed for each of these claims
reviewed in Hughes, S., Jagannath, A., Rodgers, J., Hankins, M. W., Peirson, S. N., & Foster, R. G. (2016). Signalling by melanopsin (OPN4) expressing photosensitive retinal ganglion cells. Eye (London, England), 30(2):247–254. https://doi.org/10.1038/eye.2015.264)
6.- Line 59. “Behavioral disorders” is vague, can you list them instead?
The participation of melanopsin has already been demonstrated in different behavioral disorders, such as the seasonal affective disorder [3] and delayed sleep/wake timing [4].
7.- Line 60. The link between OPN4 and ipRGCs is not explicitly stated and so the relevance is unclear to the non-expert.
melanopsin mediated phototransduction performed by the ipRGCs is complementary to the activity of cones and rods for SCN photosynchronization, but only the ipPRG axons constitute the retinohypothalaic tract that conducts the nerve impulse required for SCN photosynchronization [3]
8.-Line 62-65. Some citations are needed.
It is then understandable that a large number of investigations have focused on the relationship of melanopsin and ipPRG in behavioral disorders of the circadian rhythm [reviewed in 4].
9.- Line 69. What is known about D444G, functionally?
D444G polymorphism is a change of aspartic acid by glycine in position 444, it is frequent in African population (4.9%) and has been classified as benign variant with no associated phenotype (https://www.ncbi.nlm.nih.gov/clinvar). This information has been included in the introduction
10.- Line 70 - 73. The logic here is not well articulated. ipRGCs sense light for circadian rhythms (process C), but you are investigating insomnia (usually thought to be a disturbance of the sleep homeostat, process S). Can a more thorough argument be made? This is important because the scientific rationale for your study is not clear at the moment.
Affective and sleep disorders are related. A correct sleep homeostasis is dependent upon an efficient sensing of environmental light. The P10L SNP associated to affective disorder, could be also linked to a defective sensing of enviromental light leading to perturbance in sleep homeostasis, the worst of that being the difficulty to initiate and/or maintain this process. Hence we wanted to seek for an association between the P10L polymorphism OPN4 gene and insomnia, thus, the aim of this study was to determine whether there was an association between OPN4 gene P10L polymorphism and chronic insomnia of uncertain etiology in Mexican population.
11.- Line 159. I don’t know what type 1 and type 2 obesity is. Is it linked to diabetes status? The authors should briefly explain.
It was found in the insomnia group the trend to obesity: overweight in 44.8%, type 1 obesity defined as body mass index (BMI) between 30-34.9, in 13.8%, and type 2 obesity in 6.9% (BMI between 35-39.9)[Weir CB, Jan A. BMI Classification Percentile And Cut Off Points. [Updated 2020 Jul 10]. In: StatPearls [Internet]. Treasure Island (FL): StatPearls Publishing; 2020 Jan-. Available from: https://www.ncbi.nlm.nih.gov/books/NBK541070/
12.- Table 1. I’m not sure what the numbers in brackets denote. Also, where does the n=114 come from?
Numbers in brackets denote the percentage of individuals with the referred characteristic (apnea, sleepiness, etcetera) within each group, either control or insomnia. N=114 was a typo, now is deleted.
13.- Line 201. Doesn’t citation 27 discuss this?
If we got the point made by the reviewer, citation 27 refers to self-reported chronotypes by insomniacs, whereas our assertion goes into describing the several associations of P10L OPN4 gene SNP to sleep disorders.
Reviewer 2 Report
The authors investigated the relationship between the P10L polymorphism of the OPN4 25 gene and chronic insomnia of uncertain etiology in the Mexican population. Genetic analyzes showed that the T 29 P10L allele increased the risk of chronic insomnia in T/T or C/T individuals. The main remark concerns the small sample of tests performed. The study group consisted of 29 people and the control group consisted of 98 people. Therefore, the interpretation of the results must be careful and it should be emphasized in the manuscript that these are preliminary studies. The demonstration of a 10-fold increase in exposure to insomnia in people with a single T P10L allele and a 40-fold increase in the TT genotype may be caused by the small number of people in the study group.
Moreover, when comparing the study group with the control group (Table 1), the Bonferroni correction should be used as a method of counteracting the problem of multiple comparisons.
The authors state that about half of the surveyed people were overweight and obese. It would be good for the authors to specify what type of obesity dominated in the studied people (whether adipose tissue was accumulating in the abdominal area or trochanter). The authors should also provide the gender proportions in the test and control groups, and compare the frequency of overweight and obesity between the test and control groups.
The results would be clearer if the authors standardized the notation, e.g. they used p = <0.001 and analogically p = <0.00001 (line 225, Table 2). and reduced the number of decimal places in the results of statistical power (Table 2.).
The authors did not attempt to determine the relationship of the P10L polymorphism of the OPN4 gene with the differences between the chronotypes in the study group and the control group by means of multivariate logistic regression analysis.
Author Response
REVIEWER 2 COMMENTS
We uploaded the track changes word document : P10L_ijerph-1052002ms_VersionafterReviewersTrackChnges, this file does not follow the original line numbering, please see the file: P10L_ijerph_1052002ms_VersionafterReviewers with the changes incorporated to follow the original line numbering
1.- The authors investigated the relationship between the P10L polymorphism of the OPN4 25 gene and chronic insomnia of uncertain etiology in the Mexican population. Genetic analyzes showed that the T 29 P10L allele increased the risk of chronic insomnia in T/T or C/T individuals. The main remark concerns the small sample of tests performed. The study group consisted of 29 people and the control group consisted of 98 people. Therefore, the interpretation of the results must be careful and it should be emphasized in the manuscript that these are preliminary studies. The demonstration of a 10-fold increase in exposure to insomnia in people with a single T P10L allele and a 40-fold increase in the TT genotype may be caused by the small number of people in the study group.
Now is stated in the text that that these are preliminary results (abstract line 30; conclusions line 275), and the statistical power is now specified in the abstract too.
2.- Moreover, when comparing the study group with the control group (Table 1), the Bonferroni correction should be used as a method of counteracting the problem of multiple comparisons.
We acknowledge the problem of multiple comparisons, but we disagree that we are facing that in our study. The problem of multiple comparisons refers to test the same dependent variable in more than 2 groups, or in successive tests, whereas in our study we are comparing several dependent variables in just two groups, namely control and insomnia individuals. On the other hand, in an exploratory study, like ours, it is better not to miss a possible effect and thus avoid a Type II error and therefore not to use a Bonferroni correction. (Based on Armstrong RA. When to use the Bonferroni correction. Ophthalmic Physiol Opt 2014; 34: 502–508. doi: 10.1111/opo.12131)
3.- The authors state that about half of the surveyed people were overweight and obese. It would be good for the authors to specify what type of obesity dominated in the studied people (whether adipose tissue was accumulating in the abdominal area or trochanter).
This explanation is now included in the new version of the manuscript: “It was found in the insomnia group the trend to obesity, defined as body mass index (BMI) (Weir CB, Jan A. BMI Classification Percentile And Cut Off Points. [Updated 2020 Jul 10]. In: StatPearls Treasure Island (FL): StatPearls Publishing; 2020 Jan-. Available from: https://www.ncbi.nlm.nih.gov/books/NBK541070/].”: overweight in 44.8%, BMI between 25-29.9, type 1 obesity BMI between 30-34.9, in 13.8%, and type 2 obesity in 6.9% (BMI between 35-39.9). Whereas this distribution in control group was restricted to normal weight 59.6%, overweight 24.6% and type 1 obesity 9.7%. with no type 2 obesity. These differences were significant (Fisher´s exact test, p< 0.012)
4.- The authors should also provide the gender proportions in the test and control groups, In control group there were 56 men (56.8%) and 42 women (43.2%); in insomnia group there were 9 men (31%) and 20 women (69%), now stated in the second version of the manuscript.
and compare the frequency of overweight and obesity between the test and control groups.
This comparison is mentioned in the preceding paragraph: insomnia group: overweight in 44.8%, BMI between 25-29.9; type 1 obesity BMI between 30-34.9, in 13.8%, and type 2 obesity in 6.9% (BMI between 35-39.9). Control group: overweight 24.6% and type 1 obesity 9.7% with no type 2 obesity. These differences were significant (Fisher´s exact test, p< 0.012)
5.- The results would be clearer if the authors standardized the notation, e.g. they used p =
|
<0.001 and analogically p = <0.00001 (line 225, Table 2). and reduced the number of decimal places in the results of statistical power (Table 2.)
Now the notation for p-values has been standardized/corrected. In the results of statistical power, the decimal places have been reduced to one in both the abstract and table 2.
6.- The authors did not attempt to determine the relationship of the P10L polymorphism of the OPN4 gene with the differences between the chronotypes in the study group and the control group by means of multivariate logistic regression analysis.
We did not find correlation between genotype and chronotype in the insomnia group (line 467), neither differences between chronotypes of patients when comparing with chronotypes of controls (line 499), so it is puzzling what the reviewer is suggesting. We must say we can not answer his point.
Round 2
Reviewer 2 Report
The authors made corrections to the manuscript in line with my previous comments. The manuscript is well written and informative. In my opinion, there are no objections to the revised manuscript. The article may be adopted in this form.